

# Responses of the 14 October 2023 annular solar eclipse observed in satellite temperature profiles

Ana Roberta Paulino[1], Igo Paulino[2], and José Augusto Pereira[1]

[1]Departamento de Física, Universidade Estadual da Paraíba, Rua Baraúnas, 351, Campina Grande, PB, Brazil
[2]Unidade Acadêmica de Física, Universidade Federal de Campina Grande, Av. Aprígio Veloso, 882, Campina Grande, PB, Brazil

**Correspondence:** Ana Roberta Paulino (anaroberta.paulino@servidor.uepb.edu.br)

**Abstract.** On 14 October 2023, an annular solar eclipse occurred, in which, the umbra crossed part of the North Pacific, North, Central and South Americas ending over the South Atlantic ocean. On the day of the eclipse the instrument Sounding of the Atmosphere using Broadband Emission Radiometry (SABER) on board the Thermosphere Ionosphere Mesosphere Energetics Dynamics (TIMED) measures temperature profiles in three orbits crossing the área of the abrangency of the eclipse. One of these orbits registered almost simultaneous events to the eclipse path over the Colombia area. Temperature data from 07 to 21 october 2023 were averaged in that region within a grid of $15^o \times 15^o$ (latitude x longitude) to establish the control profile. Furthermore, comparisons to profiles observed in the days before and after were done to investigate likely instantaneous effects of the atmosphere to passage of this eclipse. The main results were changes in the temperatures producing cooling of 9-45 K in the troposphere, lower mesosphere and mesopause and warming of 7 K in the stratosphere. These results compared favourably to previous temperature observations during other eclipses and confirmed the potential of the SABER instrument to investigate transient events as solar eclipses.

## 1 Introduction

A solar eclipse occurs when the Moon is placed between the Earth and Sun, blocking the sunlighting. Depending on the distance between the Moon and Earth, the eclipse can be total (when the Moon is closer to the Earth) or annular (if the Moon is distant). One can observe a total or annular eclipse if it is within the area of the disc of the Moon's shadow (umbra) or partial eclipse if it is outside the disc area (penumbra). Although the solar eclipses being global events, they can be seen only by observers restricted to the penumbral area.

In the last decades, effects of the solar eclipses in the atmosphere have called the attention of the scientific community, primarily due to the advances in the atmospheric measurements, data processing and numerical simulation, which are tools that help to better understand the processes involved in those events.

Physically, the supersonic motion of the Moon's shadow carries the solar radiation turned off across the atmosphere. Thus, several consequences were introduced into the whole atmosphere. The main direct effects across the path of the umbra are the atmospheric cooling near the Earth's surface (e.g., Sergeeva et al., 2025), small changes of the ozone production in the stratosphere (e.g., Bernhard et al., 2025) and reduction of the ionospheric ionization (e.g., Sergeeva et al., 2025). Secondarily,



atmospheric disturbances, e.g., gravity waves (e.g., Harding et al., 2018; Paulino et al., 2020) and medium-scale travelling ionospheric disturbances (e.g., Chen et al., 2022) can be produced and they can propagate into the atmosphere producing complexes coupling processes.

On 14 October 2023, an annular solar eclipse started over the North Pacific. The Moon's shadow umbra crossed part of the United States of America, Mexico, Guatemala, Belize, Honduras, Nicaragua, Costa Rica, Panama, Colombia and Brazil. The

eclipse ended over the South Atlantic ocean. Other countries observed the partial obscuration like Canada, countries in Central and South Americas and the Caribbean.

This particular eclipse had a maximum obscuration of ∼90%, which promoted an amazing event of a 'ring of fire' crossing the sky where the annular eclipse could be observed. Additionally, the Moon's shadow of about 200 km in diameter ran through the atmosphere with a speed from 760 m/s (2736 km/h) to 3100 m/s (11160 km/h).

On the day of the eclipse, the Thermosphere Ionosphere Mesosphere Energetics Dynamics (TIMED) sounded the atmosphere using the Sounding of the Atmosphere using Broadband Emission Radiometry (SABER) instrument. The satellite crossed three times the area covered by the eclipse path and during one of these passages, there were almost coincident measurements with the Moon's shadow over the South American. The present manuscript reports those observations, which revealed salient aspects of the structures and their evolutions in the vertical temperature profiles, which were likely responses of the atmosphere to this

eclipse. As each eclipse has particular characteristics, which produce different responses of the atmosphere, the findings of this work certainly contributes to advances in understanding the changes in the atmosphere associated with the transient and rare events like solar eclipses.

## 2  SABER Measurements

The Sounding of the Atmosphere using Broadband Emission Radiometry (SABER) is a radiometer which measures radiation of

the $CO_2$. To estimate the temperature profiles it uses the local thermodynamics equilibrium in the troposphere and stratosphere and non-local thermodynamics in the mesosphere and lower thermosphere (MLT), see Mertens et al. (2001) for further details. Instrumental errors are more expressive in the MLT varying from 1.4 K at 80 km altitude to 22.5 K above 110 km (Mertens et al., 2009).

In the America sector, during the afternoon on 14 October 2023, the SABER instrument collected data in three orbits.

Figure 1 shows this configuration, in which part of the measurements of the three orbits are shown on map. Dots represent approximately the position of the sounding and their colors represent the solar local time referenced to the color bar. The time of five soundings in the central orbit were highlighted on the map. As the TIMED is an almost sun-synchronous satellite, there are not many differences between the time of the orbits.

In addition, the path of the eclipse was also projected on the map of Figure 1. Again the colors of the central line represent

the solar local time of the eclipse. One can observe that the measurements of the left orbit crossed the path of the eclipse later than the Moon's shadow. The measurements of the right orbit occurred earlier than the passage of the Moon's shadow. On the other hand, the measurements of the central orbit were almost simultaneous to the local eclipse. The almost coincident profiles

were highlighted In Figure 1 with the time of measurements. In this case, the profile around 14:10 (SLT) from the central orbit was retrieved immediately after the eclipse and it will be used as a reference to show and discuss the results of the work. Please note that the soundings were taken from the north to the south, i.e., the southern measurements were later than northern ones.

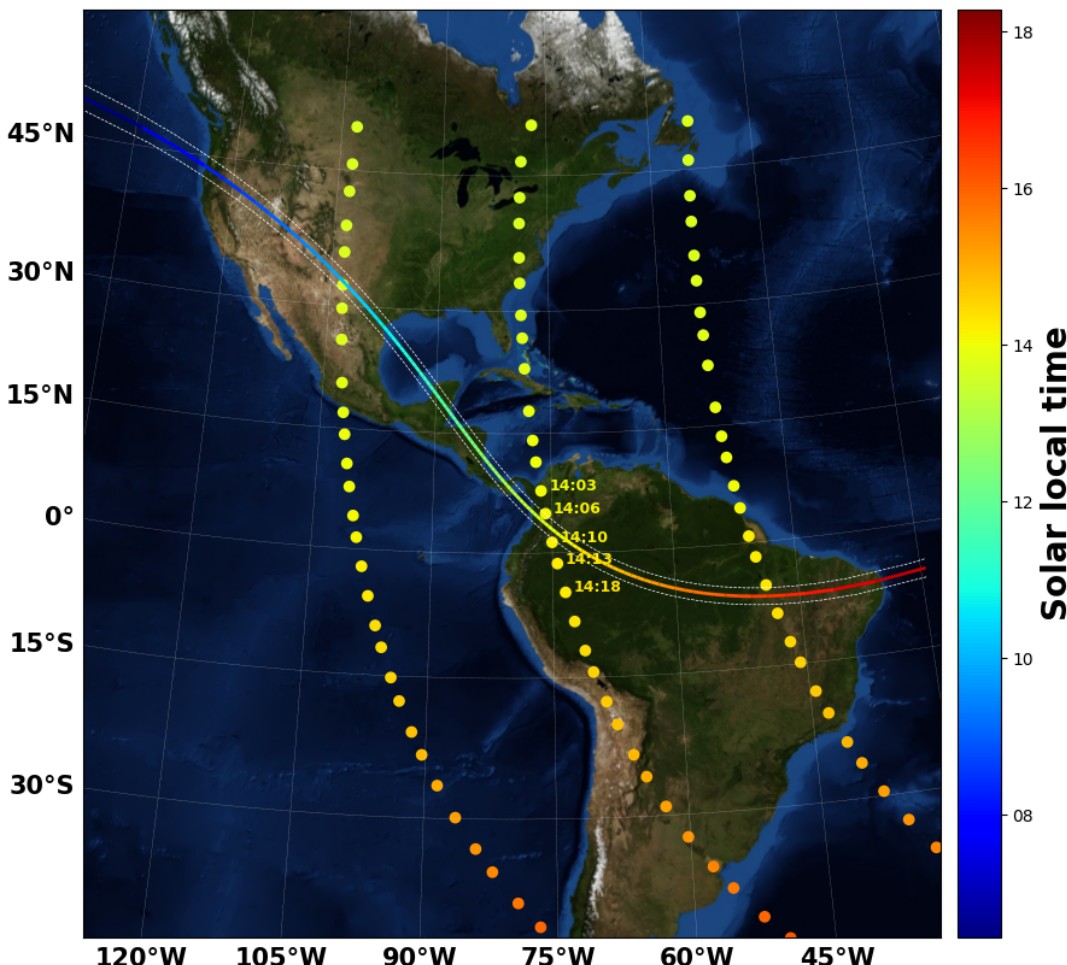

**Figure 1.** Projection of eclipse path and satellite soundings on 14 october 2025 over a map. The colors represent the solar local time referenced in the colorbar. Dashed white lines are the upper and down limits for the eclipse umbra.





## 3 Data Analysis

In order to show the likely responses of the atmosphere to the eclipse, it was necessary to know the background atmosphere near the region of interest. So, 14 days satellite measurements were used in a grid of $15^o \times 15^o$ (latitude $\times$ longitude) centered at the point in which there were almost coincident SABER measurements to the eclipse path, as described above. Figure 2 shows the positions (left panels) of the SABER soundings from 07 to 21 October 2023, excluding the day of the eclipse. As a result of these measurements, an average of the vertical temperature profile was taken as the control reference profile and it is shown on the right of Figure 2 as black heavy line. Red lines represent the individual profiles used to make the average.

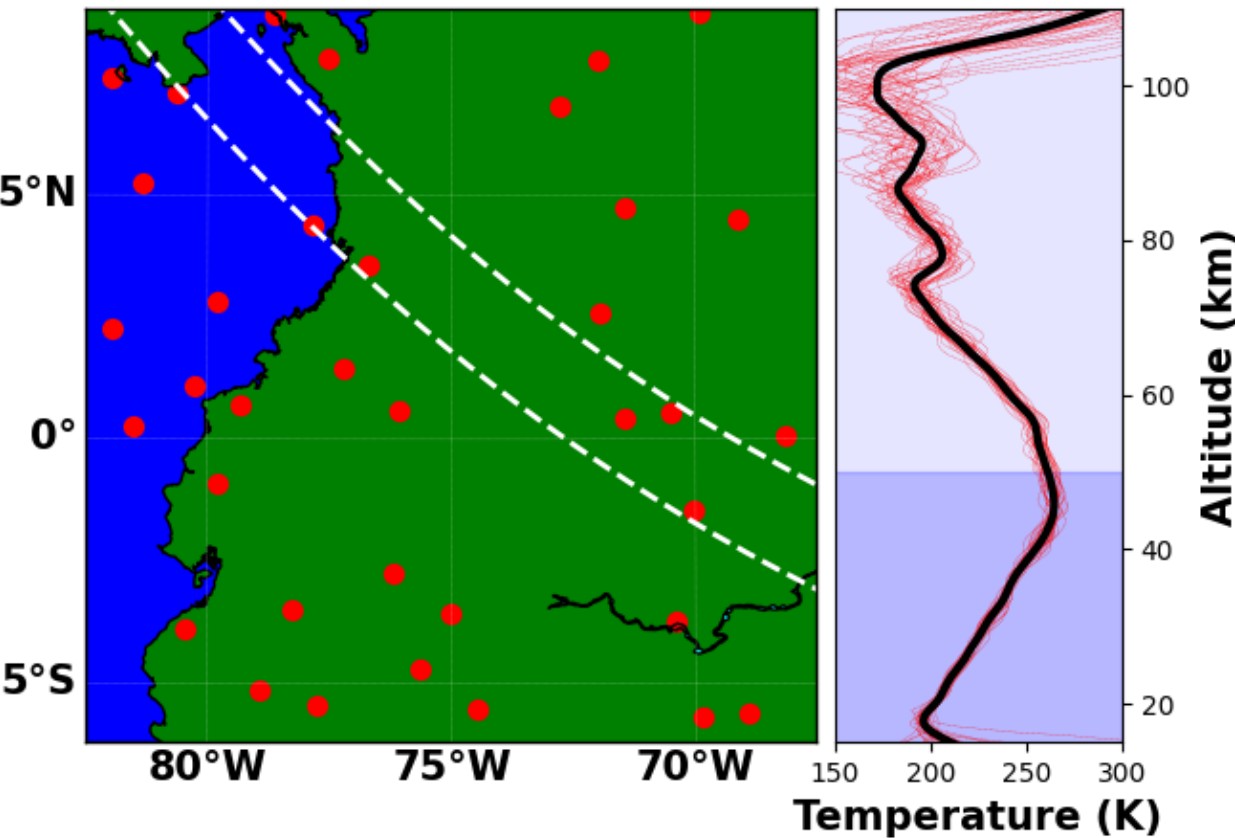

**Figure 2.** On the left, location of SABER measurements from 07 to 21 October 2023, excluding the day of the eclipse, on a grip of 15 degrees latitude times 15 degrees longitude, centered on the position of the profile almost coincident to the eclipse. On the right, temperatura vertical averaged profile (black line) using all sounding showed on the left. The read lines are all the individual profiles.





One can observe that in the lower part of the atmosphere (tropopause and stratosphere highlighted in the purple area) there is small variability in the temperature profiles. However, in the mesosphere and lower thermosphere (MLT) (light purple area), a very large variability can be observed. The large variability of the MLT region depends on several aspects with several time scale from minutes to years (Sergeeva et al., 2025), atmosphere waves have an important role and act as coupling mechanism between the atmospheric layers, then, transient events like eclipses can disturb the basic state of the atmosphere.

Another important aspect is to compare the temperature profiles to adjacent days to observe possible real effects of the eclipse. In order to do that, vertical profiles of 13 and 15 October 2023 close to the same area and same time were used. Again, as the TIMED is an almost sun-synchronous satellite, exact coincidence on time and location is not possíble.

Figure 3 shows temperature vertical profiles from 15 to 50 km, which comprises the tropopause and stratosphere. Upper panel for 13 October 2023, middle panel for 14 October 2023 and lower panel for 15 October 2023. Two regions were highlighted, one below 17 km altitude and another one between 31 and 37 km altitude.

A cooling of the tropopause was observed on 14 october 2023 starting at 14:15 (SLT) with amplitude of -9.5 K and ending at 14:34 (SLT) with amplitude 2.3 K. The maximum cooling was 9.5 K, observed at 14:15 (SLT). The cooling was considering comparing the instantaneous measurements to the average temperature (black line), in the case of the troposphere, the altitude of 15.1 km was taken as a reference because bolow of the altitude, sometimes the instrument misses points. This behavior was not observed on 13 october 2023, although a cooling of the tropopause was observed on 15 October 2023 starting before 13:55 (SLT) and ending at 14:09 (SLT).

On 14 October 2023, a warming could be observed in the stratosphere starting at 14:10 with amplitude of 3.0 K, reaching 8.9 K of amplitude at 14:23 and ending at 14:28 with amplitude of 6.2 K. This time interval is practically the same as the troposphere cooling described above. If one compares the other days, there is no similar behavior and the amplitude of the warming is quite greater than any other proeminence found in the stratosphere. There was also observed a positive vertical phase progression of 2.7 m/s from 14:18 (SLT) to 14:28 (SLT).

Returning to Figure 1, one can see that the profiles after 14:28 are getting far away from the eclipse area (longer than 1500 km) and a natural damping is expected. Before 14:10, there was no effect of the eclipse in the data, so the temporal range from 14:10 and 14:28 seems to be feasible as a likely influence of the eclipse.

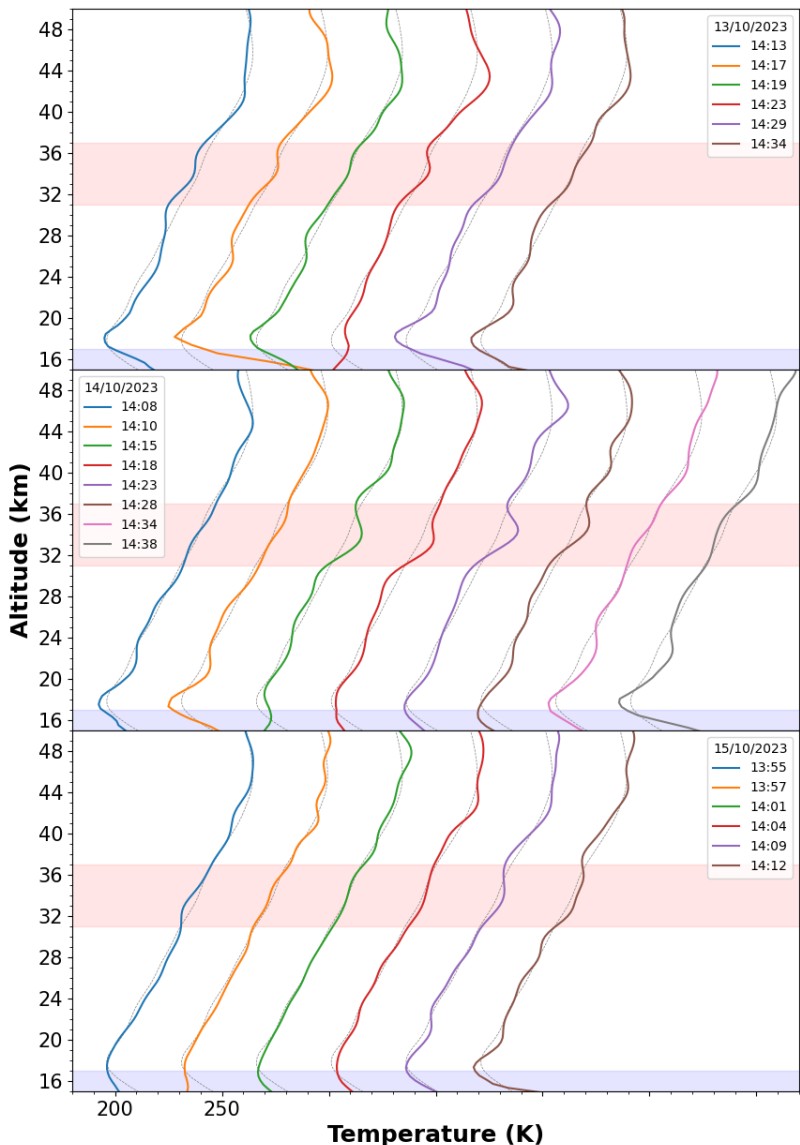

**Figure 3.** Temperature vertical profiles observed by the SABER instrument on 13 (upper panel), 14 (middle panel) and 15 (lower panel) October 2025. Two regions of each panel were highlighted: below 17 km altitude (tropopause) and between 31 and 37 km altitude (middle of the stratosphere).





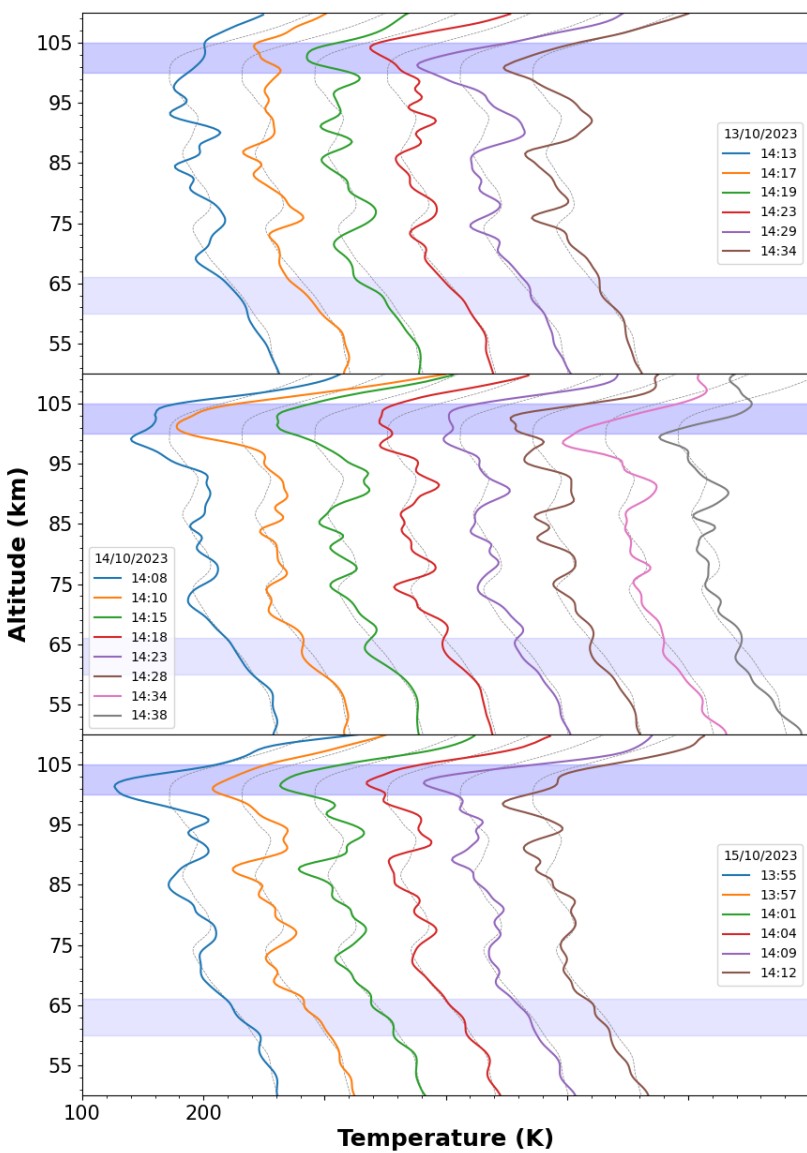

**Figure 4.** Same of Figure 3 but for the altitude range from 50 to 110 km. Again two regions were highlighted, one in the mesosphere around 64 km and another in the mesopause around 104 km altitude.



Table 1 shows all the calculations mentioned above, i.e., amplitudes, vertical position and vertical phase progression for the coolings and warmings and the horizontal distances of the satellite measurements as well.

**Table 1.** Table 1 - Amplitudes, altitudes, vertical speed and horizontal distances of the coolings and warmings.

| | SLT | 14:08 | 14:10 | 14:15 | 14:18 | 14:23 | 14:28 | 14:34 | 14:38 |
|---|---|---|---|---|---|---|---|---|---|
| Tropopause | **Altitude (km)** | 15.1 | 15.1 | 15.1 | 15.1 | 15.1 | 15.1 | 15.1 | 15.1 |
| | **Amplitude (K)** | -5.6 | 2.4 | -9.5 | -8.0 | -5.9 | -8.7 | -2.3 | 16.0 |
| | **Vertical speed (m/s)** | – | – | – | – | – | – | – | – |
| Stratosphere | **Altitude (km)** | 34.0 | 35.5 | 33.6 | 32.2 | 34.0 | 34.8 | 32.9 | 34.4 |
| | **Amplitude (K)** | -0.7 | 3.0 | 6.7 | 6.4 | 8.9 | 6.2 | -3.5 | -2.9 |
| | **Vertical speed (m/s)** | – | – | – | 2.7 | 2.7 | 2.7 | – | – |
| Mesosphere | **Altitude (km)** | 62.5 | 62.5 | 63.7 | 63.3 | 62.9 | 62.6 | 62.9 | 62.2 |
| | **Amplitude (K)** | -2.4 | -9.6 | -13.5 | -9.5 | -8.7 | -10.9 | -10.4 | -14.1 |
| | **Vertical speed (m/s)** | – | – | -2.2 | -1.9 | -1.2 | -1.0 | – | – |
| Mesopause | **Altitude (km)** | 104.5 | 103.4 | 104.2 | 105.6 | 104.2 | 102.7 | 100.6 | 99.5 |
| | **Amplitude (K)** | -38.0 | -53.9 | -45.8 | -34.7 | -30.2 | -26.1 | -21.2 | -15.5 |
| | **Vertical speed (m/s)** | – | – | – | -4.7 | -4.9 | -5.4 | -5.1 | -4.6 |
| | **Horizontal distance from nearest point (km)** | 248.6 | 0.0 | 478.6 | 705.9 | 1183.1 | 1517.3 | 1993.0 | 2213.8 |

## 4 Discussion

Several observational experiments have been performed to provide data that can contribute to advances in understanding how the atmosphere responds to the passage of experiments. In particular, some works have been published about the annular solar eclipse of 14 October 2023, including satellite and ground based observation. For instance, Aryal et al. (2025) used data from the GOLD satellite and showed a reduction of $\sim$100 K in the thermospheric temperature, which is one of the most expressive responses of the atmosphere to the passage of this eclipse. Furthermore, Das et al. (2024) used multi-instruments to investigate the total electron content and critical frequency of the $F_2$ ionospheric layer. Sergeeva et al. (2025) also investigated the effects of this eclipse in a comprehensive multi-instrument work covering several atmosphere levels including the surface measurements. Bernhard et al. (2025) using data collected during three solar eclipses, including the 14 october 2023 eclipse, and point out a reflection about the real enhancement of the ozone concentration in the stratosphere during the passage of eclipses.

Other recent annular solar eclipses have also been investigated in different atmospheric levels (e.g., Ratnam et al., 2011; Subrahmanyam et al., 2011; Ravindra Babu et al., 2022; Prakash et al., 2022; Das et al., 2023; Choudhary et al., 2024). As each eclipse is a unique geophysical event, different experiments and measurements are important to understand the atmosphere





under these special conditions. In the present work, the coincident SABER measurements to the eclipse path give some different
insights related to the nature of the observation and the temporal and spatial covering of the measurements.

The present results show the complexity of the radiance balance which the atmosphere experienced by the passage of the
annular eclipse on 14 October 2023. The first important result was the decrease of the temperature in the troposphere of ∼3-9.5
K. Previous observational reports of solar eclipse effects in the atmosphere showed decreases of the troposphere temperature
with similar amplitudes (e.g., Das et al., 2023; Subrahmanyam et al., 2011). Even the 14 October 2023 being a annular eclipse,
its effects were relatively expressive in amplitude compared with previous observations.

Although the solar eclipse be a transient event, it has been observed that the blocking of the sunlight during total, annular
or partial eclipse can produce cooling on the atmosphere near surface of few degrees (e.g., Chernogor, 2008; Banerjee and
Bhattacharya, 2022; Ravindra Babu et al., 2022; Prakash et al., 2022; Sun et al., 2022; Elmhamdi et al., 2024, and references
therein). However, the fluctuation along the troposphere and high levels could not be linear as shown by (Subrahmanyam et al.,
2011), i.e., there are different amplitudes for different altitude levels.

Maybe the most interesting result presented in this paper is the enhancement of the stratosphere temperature, which had
a well defined response reaching ∼9 K of amplitude around 34 km altitude. The observations have shown that the local
stratosphere warms and increases the ozone concentration, primarily in the lower stratosphere (e.g., Randhawa et al., 1970;
Subrahmanyam et al., 2011; Wang and Liu, 2010; Ratnam et al., 2011; Wang et al., 2012; Das et al., 2023, and references
therein) during the passage of eclipses as a direct response to the process of compression and expansion of the atmosphere. In
Figure 3 a small enhancement of the temperature in the lower stratosphere (from 20 to 204 km altitude) on the day of the eclipse
was also observed, however it was not well synchronized to the eclipse passage as it was observed in the middle stratosphere.
Besides, random enhancements in the days after and before the eclipse could also be seen. Thus, it is not easy to make a direct
association of this enhancement to the solar eclipse.

The warming in the middle stratosphere was poorly explored in the previous publication. The reason can be the historical
measurement by launching rockets and balloons, which have altitudinal limitations (e.g., Ballard et al., 1969; Randhawa et al.,
1970; Das et al., 2023, and references therein). In contrast, it is important to mention that Wang et al. (2012) showed results of
middle stratospheric warms using satellite data for a set of eclipses from 2006 to 2010. Further observations are necessary to
better understand the physical reason for this pronounced warming in the middle stratosphere and maybe numerical simulation
can help to understand whether these observations are either consequences of the geographical position of the observation near
the equator or if it is common in other regions of the planet. What calls the attention is that the amplitude of the warm has the
same magnitude of the tropospheric cooling.

In the mesosphere and lower thermosphere, the dynamic is quite complex due to the presence of large amplitude tidal,
planetary and gravity waves and interaction among them. A cooling greater than 10 K was observed on 14 October 2023
compared with the control profile and the days after and before. This cooling was observed around 63 km altitude and had
temporal evolution similar to the warming in the middle stratosphere. As the MLT is remote and difficult to make observations,
it has been poorly explored during solar eclipses. Even so, some reports were done, for instance, Schmidlin and Olsen (1984)



observed a cooling of ∼10 K above 52 km altitude on the 26 February 1979 solar eclipse. Ballard et al. (1969) also reported a decrease of the temperature between 50 and 60 km altitude during the passage of the 12 November 1966 solar eclipse.

In the mesopause altitudes, there was also observed a significant cooling on 14 october 2023 which started with amplitude over 50 K and decreased with the time and distance from the Moon's shadow path. The day after the eclipse also showed a moderate cooling around the same time, but without expressive vanishing along the time. That is another region of the atmosphere with few reports, for instance, Sumod et al. (2011) presented results for the 15 January 2010 eclipse with tendency of cooling in this region, but it was not the focus of that article.

In the thermosphere, the temperature fluctuations might be higher amplitudes because the absorption of energy has a direct response increasing the temperature. Observations have shown cooling of order of 100 K (e.g., Aryal et al., 2025) and (McInerney et al., 2018) predicted changes of ∼ 40 K in the thermosphere for the 21 August 2017 Solar Eclipse. In addition, using the daily averaged profile, Barad et al. (2022) showed a decrease of ∼11 K in the temperature profile around 98 km altitude on 26 December 2019.

The results presented in this paper confirm the relevance of the SABER measurements to investigate transient events as solar eclipses. The solar eclipses that can occur in the middle and lower latitudes have a large probability to find coincident measurements that could support and corroborate several experiments and observations.

## 5 Conclusions

On 14 October 2023 occured an annular solar eclipse, which crossed part of the Pacific ocean, North, Central and South America and ended on the Atlantic ocean. The scope of this paper was to describe likely effects of the eclipse in the temperature profile from the upper troposphere to the lower thermosphere. In order to do that, temperature profiles retrieved by the SABER instrument on board the TIMED satellite were used for a coincident orbit which crossed almost simultaneously the path of the Moon's shadow. Profiles from 07 to 21 October 2023 within a box of $15^o \times 15^o$ (latitude and longitude) were averaged to serve as control prifle and the profiles in the same área in days after and before were used to make comparisons. The main findings were:

- A cooling of ∼9 K in the troposphere at 15.1 km altitude;

- A warming of ∼7 K in the stratosphere around 33 km altitude;

- A cooling >10 K in the mesosphere around 62 km altitude;

- A cooling >45 K in the mesopause around 104 km altitude.

Comparison to previous observations during other total annular or partial eclipses showed compatibility in values of the cooling and warming in the troposphere, stratosphere and mesosphere. The temporal and spatial evolution of that prominences in the profiles have consistent pattern as the satellite moves away from the eclipe trail. Thus, the present results might be a direct response of the atmosphere to the passage of the eclipse.



*Data availability.* SABER measurements can be freely accessed in the Internet on https://saber.gats-inc.com/data_services.php

*Author contributions.* ARP - Conception, data analysis and writing; IP - Conception, data analysis and revision; JAP - Data analysis and
175  revision

*Competing interests.* At least one of the (co-)authors is a member of the editorial board of Annales Geophysicae.

*Acknowledgements.* The authors have been financed by the "Conselho Nacional de Desenvolvimento Científico e tecnológico (CNPq)"
under contracts 309981/2023-9, 303115/2025-4, 404971/2021-0 and Universidade Estadual da Paraiba (PRPGP no 03/2024 - Edital Bolsa
de Produtividade em Pesquisa da UEPB). The authors are grateful to the SABER team for the access to the data on http://saber.gats-inc.com.



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
