# Peer review of "Responses of the 14 October 2023 annular solar eclipse observed in satellite temperature profiles"

_EGUsphere, 2025_

## Referee Comment (RC1)

Review of the paper entitled "Responses of the 14 October 2023 annular solar eclipse observed in satellite temperature profiles" submitted to EGUsphere by A. R. Paulino et al.

**Summary**

This paper describes observations of temperature in the region from 15 km to 105 km as measured by the SABER instrument on the TIMED satellite during the annular solar eclipse event that occurred on 14 October 2023. The observations are potentially of interest to see how the Earth's middle atmosphere responds to transient events. The authors examine SABER temperatures prior to and during the eclipse and from these observations determine the magnitude of the temperature change induced by the eclipse.

**Recommendation**

Regretfully, the paper must be rejected, but not for any fault in the analysis by the authors, but because the SABER temperature data are not suitable for analysis during eclipse events.

Specifically, the SABER temperature algorithm is not designed for, and does not account for, conditions during the eclipse. As described in the papers by Mertens et al. referenced by the authors, the SABER temperature retrieval involves complex non-LTE radiative transfer calculations involving the vibration-rotation bands of carbon dioxide ($CO_2$) in the 15-micrometer spectral region. The non-LTE processes are substantially different for day than for night in that there is substantial absorption of solar radiation by $CO_2$ during the day and of course, none at night. Consequently, the vibrational temperatures of the 15 micrometer bands of $CO_2$ have a strong variation from day to night. To derive the temperature correctly, the SABER radiative transfer models must first compute the correct vibrational temperatures.

The algorithm used to analyze the SABER data during the eclipse is the daytime algorithm. It is not possible to switch from day to night for one or two profiles during operational processing. In addition, the SABER team has examined several eclipse conditions and found that even when near or "in" the eclipse region, the atmosphere that SABER views is almost always partly illuminated, so it is never completely in night conditions. For this reason, there is really no way for SABER to derive a valid temperature profile in or near to the eclipse region. (Note that SABER views the earth's limb, not in the nadir, and consequently measures infrared emission over a long (~ 1000 km) path).

Perhaps the authors might have suspected something given the magnitudes of the changes in temperature reported in their paper. For example, the 45 K decrease in 104 km might have come across as likely non-physical. Does this result mean at night the temperature would decrease by over 100 K in maybe five to ten minutes after sunset? The authors are referred to the paper by Huang et al., 2006, specifically figures 1, 2, 3, and 4.

https://agupubs.onlinelibrary.wiley.com/doi/full/10.1029/2005JA011426

These figures show the diurnal variation of the temperature at 55 km and 95 km measured by SABER and by the Microwave Limb Sounder (MLS) on the Aura satellite. The eclipse change in temperature in a couple of minutes surely cannot be larger than the diurnal change.

Also, it is quite likely that the warm stratosphere and the "troposphere" results at 15 km reported in the paper are likely algorithm effects due to the incorrect temperatures in the mesosphere and lower thermosphere arising from the daytime temperature algorithm being used instead of the night algorithm.

Lastly, there is one other point to make regarding the SABER algorithms, for the altitudes above 80 km, which again relates to the non-LTE radiative transfer calculations. A critical input to the temperature algorithm (which is discussed in the Mertens et al. papers) is the atomic oxygen concentration. SABER's temperature algorithm uses atomic oxygen provided by the MSIS-2000 empirical model. As such, the atomic oxygen concentration from the empirical model cannot be expected to be correct if there are rapidly changing conditions to be properly modeled. This fact, in addition to the daytime/nighttime issues mentioned above, likely leads to the large and almost surely incorrect temperature changes reported in the paper.

In closing, the authors are encouraged to reach out to the SABER team if they have further questions about the validity and utility of the data during transient events.

---

## Referee Comment (RC2)

Comments on the manuscript "*Responses of the 14 October 2023 annular solar eclipse observed in satellite temperature profiles*" by *A.R.Paulino, I.Paulino and J.A.Pereira*.

The study discusses a very interesting topic of temperature variations at different heights of the atmosphere during the solar eclipse. It is not ready for publication in its present form according to the comments provided below. I would recommend it for publication after a major revision.

**General comments.**

**1**. The text is not easy to read because of many sentences that provide an ambiguity of interpretations. I strongly recommend revising all the manuscript with a critical eye. Please use the accepted scientific terms introducing them at first mentioning. Please be more specific about every detail you provide.
I recommend paying attention to the formulation of the sentences and avoiding a plain language. Please see my specific comments.

**2**. Introduction section.
Please state clearly the aim of this study and the tasks to be resolved. Please avoid putting the results in Introduction (lines 38-40).

**3**. Reference values.
- An explanation is needed, why exactly 14 days of data (some of which were before and some after the eclipse) were chosen to reveal the regular tendencies?
- Is there any systematic variation corresponding to 14 days interval, which should be taken into account?
- These days are not "quiet reference days". I see at least weak and moderate magnetic field disturbances during this period. My recommendation is to check any other natural events that could perturb the atmosphere during this period.
- Why the average values are taken as a quiet reference? I would probably incline at least to a median value as, in contrast to the average, it is not affected by disturbances.

**4**. The moments of temperature decrease/increase should be corresponded to the eclipse parameters. At which phase of the eclipse the particular effect was observed? Or was it before or after? Why?

**5**. Lines 127-128: "random enhancements in the days after and before the eclipse could also be seen. Thus, it is not easy to make a direct association of this enhancement to the solar eclipse."
   I disagree.
Why the authors think that the enhancements on other days were random?
Did you check regular temperature variations in the region of the study? What are they? What is the limit of day-to-day variability in this season?
What are the possible reasons for enhancements? Can we check and discard some of them?
It is possible to associate some irregular atmospheric behaviour during the eclipse while knowing regular variations and the particular agents that can induce some temperature changes.

**6**. Line 129: "middle stratosphere was poorly explored in the previous publication" – I am not sure what previous work is referred to.
Why it was not studied in the present work at least with the same SABER instrument or may be some model? The authors stress the need to do it in lines 133-136.

**7**. Conclusions.

The last sentence introduces uncertainty: were the temperature disturbances triggered by the eclipse passage or not?

The discussion should be developed:

- What says in favor to the positive response to this question?
- What is the physical mechanism? Is it only the decrease in radiation from the Sun or some other factors? Is the physics of temperature change is the same at different heights?
- Do the rates of change correspond somehow to the eclipse parameters at different heights (obscuration rate, the speed with which the shadow passes, the direction of movement of the shadow in relation to the movement of the satellite or any other factor)?
- Do the season and the local time play some role in the middle atmosphere response to the eclipse?
- What new features are added to current understanding of temperature variations or what known features were confirmed/précised by the obtained results?

**Specific comments (examples).**

49: "during the afternoon" – Please be more specific. Responses to solar eclipse can be as fast as a minute. I suggest providing the exact hours and minutes.

51: Why approximately? I am not sure I understand the meaning of "position of the sounding". Do you mean satellite position projection?

52: Please mention here why "the time of five soundings in the central orbit were highlighted on the map"? Are these the only moments when the satellite position coincided in time and space with the eclipse penumbra?

53: "differences between the time of the orbits" – Do you mean the change of the moment of passing over the same location from day to day?

54: The details are needed. What path exactly was projected? Do the authors refer to the annular eclipse conditions or to some particular obscuration rate? Is the position of this path corresponds to the SABER orbit height?

56: The details are needed: What shadow exactly is meant (obscuration rate, height)?

57: What is meant by "almost simultaneous"?  "The almost coincident profiles" of what? I do not see profiles in Figure 1.

59: What the "immediately after the eclipse" means? Did the whole duration of the annular and partial eclipse ended at the moment of observation? If not, then what obscuration rate covered this point? How many seconds/minutes passed after the eclipse ended?

59: Please reformulate, as usually the regular variations without any perturbation are considered "a reference".

Figure 1.
Please eliminate "over a map".
"upper and down limits for the eclipse umbra "– I am not sure I understand. There was no "umbra" as the eclipse was not total. If the authors refer to the maximal phase of the eclipse, then the limits seem to be very wide and for the partial eclipse - very small. I would also recommend checking the coordinates of this path as it seems shifted from the actual central line of the eclipse.

74: close to the same area and same time were used – Please provide the exact detail: how close (in km? in grados?), how exactly the same (in minutes?) ?

Figure 3.
I see the weak grey profiles behind each color profile. Please explain what are they? Is this a reference profile for this particular moment (or some interval) ? All grey profiles seems the same to me.

I do not understand the sentence between lines 81-83. Please reformulate.

Lines 80-85: The text is confusing at the same time of the eclipse day both cooling and warming is reported.

89: Please explain what is "a positive vertical phase progression of 2.7 m/s"

90:Please reformulate: "are getting far away from the eclipse area"

92: "feasible as a likely influence "– It is to much doubt in this sentence. The other causes of temperature change should be revised to make the conclusion.

Please explain how the regions are chosen in Figures 3 and 4.

Table 1 caption:
Calculations were not mentioned. Measurement results were. Do you mean the absolute change of temperatures in K by "amplitude"?
The terms "vertical position and vertical phase progression" and "Vertical speed (m/s)" should be introduced / explained.

109: "some different insights related to the nature of the observation" – What is different? To what nature do you refer?

110: "the complexity of the radiance balance" – I do not understand what it is.

111: "The first important result was " – Just say "results", as the results are not divided in important and not important.

113: amplitudes of what? Do you mean the absolute deviation of the temperature?

114: "Even the 14 October 2023 being a annular eclipse,"
"its effects were relatively expressive in amplitude compared with previous observations"
Why Even?
What "relatively expressive" means?
Compared to other eclipses or compared to the results of other authors?

118: "However, the fluctuation along the troposphere and high levels"
Please eliminate "However" as there is no contradiction here.
Fluctuation refers to changes with time. Here, probably, different temperature behavior with height is meant.
"Along" is not a word here, as it refers to some "horizontal" path, but here the vertical profile changes are discussed.
Do you mean "stratosphere" by high level?
Please reformulate carefully.

121: "enhancement of the stratosphere temperature, which had a well defined response reaching   9 K of amplitude around 34 km altitude"
Enhancement cannot respond, the temperature can.
Response of amplitude?
Around 34 – please be specific in indicating values or ranges.

122: "The observations have shown that the local stratosphere warms and increases the ozone concentration, primarily in the lower stratosphere."
Is it a verified result of this work or just the conclusion from the literature? Please provide proofs that it was what occurred in this case.

124: "process of compression and expansion of the atmosphere."
No discussion about that was given in the text. Please mention how this compression and expansion (of what?) is triggered by the eclipse passage.

137: "In the mesosphere and lower thermosphere" - Please be more specific indicating the particular heights or height ranges, because the mesosphere and lower thermosphere is a very large height range with different dynamics. Only a height of 63 km was mentioned.

140-143: I recommend withdrawing the sentence about how difficult it is to study (here and throughout the text). The works on other eclipses may be mentioned in the Introduction section.

144: "In the mesosphere and lower thermosphere" - Please be more specific at what altitudes exactly.

146: What "expressive vanishing" means?

149-153: It is difficult to the reader to conclude if the results of different eclipses are in accord, because there is no details on the obscuration rate, direction and speed of the shadow movement, the season and local hour of the event, etc. The results under different conditions are not comparable.

171: do you mean "effects" by the word "prominences" ?
What "consistent pattern" means?
"the satellite moves away from the eclipe trail" - What is the eclipse trail exactly? What are the directions and speeds of change of the position of the satellite and the eclipse shadow?

**Minor comments.**

Please eliminate the words "One can observe / one can see" throughout the text.

Lines 28-31: This is unnecessary information is given in a plain language. The list of the countries is not significant, but the latitude (high/mid/low) and longitude sectors matter. I recommend withdrawing this paragraph. The map with the eclipse projection will do.

SABER abbreviation is introduced more than once.

49: America sector→ American sector

55: "left orbit" and "right orbit" are not the adequate terms.

58: SLT abbreviation was not introduced

60: Please eliminate the sentence as it is too obvious considering the time of measurements are given along the satellite passage projection in Figure 1.

62: "In order to show the likely responses" → To study the atmospheric response to solar eclipse "near the region of interest" → in the region.

73: Please eliminate "possible real"

97: The atmosphere does not respond to "the passage of experiments".

100: expressive responses → significant?
Why this particular feature is more important than other revealed features of this eclipse?

104: reflection about the real enhancement – I do not understand what it is.

96-110: My suggestion is to move this paragraph to Introduction section:
mentioning the works already published on the chosen eclipse event, summarizing what was found and, then, formulate the aim of the present work which bring new information to complement the already known.

115: "Blocking" is not a word here.

120: "Maybe the most interesting result presented in this paper is" – The sentence is poorly formulated. Why the doubt? Just say "We revealed that…"

125: Probably the typo here: from 20 to 204 km altitude

149-150: Please rephrase. Why might? I do not understand what responded to what.

169: "compatibility" is not a word here. What was compared to what exactly?